# Multiscale Hidden Markov Models For Covariance Prediction

## Abstract

This paper presents a novel variant of hierarchical hidden Markov models (HMMs), the multiscale hidden Markov model (MSHMM), and an associated spectral estimation and prediction scheme that is consistent, finds global optima, and is computationally efficient. Our MSHMM is a generative model of multiple HMMs evolving at different rates where the observation is a result of the additive emissions of the HMMs. While estimation is relatively straightforward, prediction for the MSHMM poses a unique challenge, which we address in this paper. Further, we show that spectral estimation of the MSHMM outperforms standard methods of predicting the asset covariance of stock prices, a widely addressed problem that is multiscale, non-stationary, and requires processing huge amounts of data.

## 1 Introduction

Hidden Markov models (HMMs) are a critical tool in many fields, including speech recognition (Huang et al., 1990), robotics (Thrun et al., 1998), and natural language processing (NLP) (Rodu et al., 2013). Spectral estimation of HMMs, used in place of the expectation maximization (EM) algorithm or Gibbs sampling, now allows for fast estimation, making HMMs feasible for large datasets (Hsu et al., 2012; Foster et al., 2012). However, modeling complex structure with a standard HMM, if even possible, can require a prohibitively large state space. Previous work using HMMs to predict volatility fail to capture multiple timescales, and additionally are not able to scale to large datasets because they are estimated by EM (Rossi & Gallo, 2006; Nystrup et al., 2016). To specifically address datasets with multiple timescales, we propose the multiscale hidden Markov model (MSHMM), which utilizes computationally efficient spectral estimation to accommodate large, high-dimensional datasets, and can be usefully applied to multiscale non-stationary problems of industrial scale. However, variants of FHMMs have been proposed for multiscale tasks such as speech recognition, natural language modeling, and energy optimization Cetin & Ostendorf (2004); Kolter & Jaakkola (2012); Aarno & Kragic (2007); Nepal & Yates (2013); Oliver et al. (2004).

In order to motivate our method we apply the MSHMM to high-frequency asset pricing data. For US equities, sampling once per second yields roughly 5 million observations per year per stock, on thousands of stocks. Prediction of asset covariance has been recognized as an extremely important task (Wu & Xiao, 2012), and continues to be of interest (Nystrup et al., 2017), but modeling at that scale has proven challenging and often intractable. Since asset covariance is a non-stationary stochastic process, the MSHMM structure is well suited for this problem.

Our main contribution is a novel variant of multiscale hidden Markov model that uses a unique, structured hierarchical form to force the model to evolve at multiple timescales. **The prediction phase of MSHMM differs markedly from other hierarchical HMMs due to the additive emission and multiscale noise structure.** The resulting MSHMM provides a model that allows for spectral estimation, thus avoiding computationally costly EM or Gibbs sampling. Spectral estimation's ability to offer additional speed, while avoiding local minima, allows us to use our method to model large-scale multivariate time series such as high-frequency stock trades. We prove that MSHMM is consistent under spectral estimation, and we provide an efficient estimation algorithm. We show that MSHMM can achieve excellent performance in the real world task of predicting realized covariance and compare to existing methods.

## 2 MULTISCALE HIDDEN MARKOV MODEL

A hidden Markov model characterizes a sequence of discrete-time observations (in this paper we consider high-dimensional, continuously distributed observations) as being emitted by a chain of discrete hidden states. There are two main assumptions, 1) that observations are conditionally independent given the hidden states, and 2) that the hidden states are governed by a Markov process, meaning that the distribution over hidden states at time $t$ given the entire latent state history depends only on the previous hidden state at time $t - 1$, so

$$\Pr[h_t \mid h_{t-1}, \ldots, h_1] = \Pr[h_t \mid h_{t-1}].$$

We can fully specify the HMM by a transition matrix $T \in \mathbb{R}^{m \times m}$ where $m$ is the number of hidden states, an emission probability $\lambda$ that specifies the conditional probability of an observation $x$ given a hidden state $h$, and an initial distribution $\pi$ over hidden states.

$$\pi_i = \Pr[h_1 = i], \qquad T_{i,j} = \Pr[h_{t+1} = i \mid h_t = j], \qquad [\lambda(x)]_i = \Pr[x_t \mid h_t = i].$$

It is convenient to think of hidden states as being an indicator vector in $\mathbb{R}^m$, and $\lambda$ as associating with each observation $x$ a vector $\lambda(x)$ where $\lambda(x)_i = \Pr[x \mid h = i]$. For readers familiar with HMMs, $x$ is often discrete, allowing $\lambda$ to be easily interpreted as a matrix with finitely many rows. As previously mentioned, we assume our observations are high-dimensional $x \in \mathbb{R}^n$, and we assume that $m \ll n$.

HMMs can model many behaviors exhibited by data. Flexibility is obtained by increasing the size of the hidden state space, at the cost of an often prohibitive increase in sample complexity. This can be avoided by factoring the hidden state space into multiple layers and placing a nontrivial dependence structure on these layers and emitted observations (Ghahramani & Jordan, 1997), known as a factorial HMM (FHMM). Figure 1 (a) displays the dependence structure of the original FHMM.

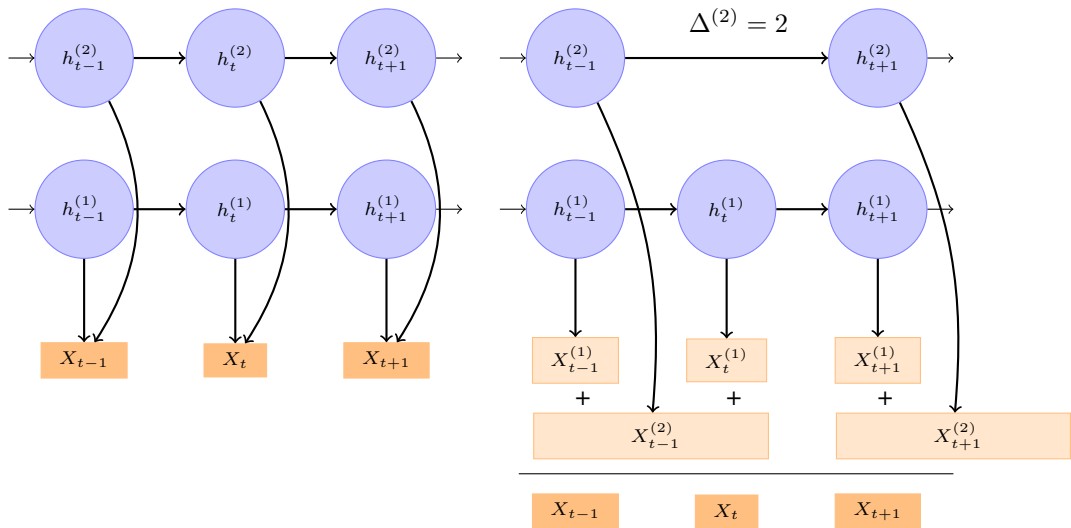

Figure 1: (a) Factorial HMM with 2 hidden state layers. (b) An MSHMM with two HMMs where $h_t^{(2)}$ is the hidden state sequence of the slower moving HMM with transitions happening at time steps $\Delta^{(2)} = 2$. The observations $X_t$ are the sum of the emissions of the two HMMs, so $X_t = X_t^{(1)} + X_{t-1}^{(2)}$.

In contrast to the FHMM, MSHMM has multiple, independently evolving HMMs with additive emissions as seen in Figure 1 (b). Consider a system with $M$ hidden state chains $\{h^{(1)}, \ldots, h^{(M)}\}$. Each hidden state chain $h^{(i)}$ transitions at different time periods $\Delta^{(i)}$, so that at a given time $t$, chains transition by time $t + 1$ only if they satisfy $(t + 1 \text{ modulo } \Delta^{(i)}) = 0$, with self-transitions allowed. In this paper, we assume that the parameters $\{\Delta^{(1)}, \ldots, \Delta^{(M)}\}$ are unique, and thus without loss of generality ordered from smallest to largest, so that the hidden state chain $h^{(M)}$ is the slowest evolving chain. Unlike in Ghahramani & Jordan (1997), where observations are sampled from a

Gaussian centered at a linear combination of the hidden states of each chain, in our model each hidden state chain emits an unobserved output *with noise*, and the system-wide observations are linear combinations of the output. In other words, we can think of each layer as an actual HMM with unobserved, though otherwise typical, output. Critically, we assume the noise terms are resampled *only* when there is a (possibly self-) transition.

## 2.1 Specification of Multiscale Hidden Markov Model

Recalling that an HMM can be fully specified with parameters $\pi$, $T$, and $\lambda(x)$, we denote the HMM parameters of each hidden process $i$ with a superscript $(i)$, so $\pi^{(i)}$, $T^{(i)}$, and $\lambda^{(i)}(x)$. As above, $h^{(i)}$ represents the hidden state chain for hidden process $i$, and $h_t^{(i)}$ its value at time $t$. Furthermore, $m^{(i)}$ denotes the size of the hidden state space for process $i$. Letting $h_t$ without the superscript $i$ denote the global hidden state of the entire system, we can characterize the independence between hidden states as

$$\Pr[h_{t+1} \mid h_t] = \prod_{i=1}^{M} \Pr[h_{t+1}^{(i)} \mid h_t^{(i)}].$$

Denote the expectation of the continuous emission for model $i$ at time $t$,

$$q(h_t^{(i)}) = E[X_t^{(i)}|h_t^{(i)}]$$

and let $q(h_t)$ be a matrix collecting $q(h_t^{(i)})$ on the columns, so

$$q(h_t) = \left[ q(h_t^{(1)}), q(h_t^{(2)}), \cdots, q(h_t^{(M)}) \right].$$

To incorporate noise at each level into our model, we define a matrix $\tilde{q}(h_t)$ as

$$\tilde{q}(h_t) = \left[ q(h_t^{(1)}) + \epsilon_t^{(1)}, \cdots, q(h_t^{(M)}) + \epsilon_t^{(M)} \right],$$

where $\epsilon_t^{(i)} \sim N(0, \sigma^{(i)})$; however, the model is robust to other distributions. The observable emission is then $X_t = \tilde{q}(h_t)w$ where $w$ is a vector of size $M$. Figure 1 displays the representation of MSHMM. Note that, as a consequence of the stickiness property in our model,

$$\Pr[q(h_{t+1}^{(i)}) + \epsilon_{t+1}^{(i)} = q(h_t^{(i)}) + \epsilon_t^{(i)}] = 1 \text{ if } (t+1 \text{ modulo } \Delta^{(i)}) \neq 0.$$

Finally, we let $\tilde{h}_t^{(i)} = E[h_t^{(i)}|X_{t-1} \ldots X_1]$ denote the *belief state* of HMM $i$ at time $t$.

## 2.2 Prediction Multiscale Hidden Markov Model

The prediction procedure for our model differs substantially from standard hierarchical and factorial HMMs. Typically for HMMs and variants of HMMs, it is only necessary to estimate the belief state $\tilde{h}_t$. But for the MSHMM, the correction term

$$\tilde{\epsilon}_t^{(i)} \triangleq q(h_t^{(i)}) + \epsilon_t^{(i)} - q(\tilde{h}_t^{(i)})$$

from HMM $i$ acts as persistent bias for HMM $i-1$ for a period $\Delta^{(i)}$. Therefore, in order to eliminate this bias, we must estimate $\tilde{\epsilon}_t^{(i)}$.

We require one more definition in order to specify the prediction procedure. Let $g^{(i)}(t)$ to be a step function for the $i$th hidden state chain that steps when HMM process $i$ has a transition. For example, if $t = \{0, 1, 2, 3, 4, 5, 6, \cdots\}$, and $\Delta^{(3)} = 3$ then $g^{(3)}(t) = \{0, 0, 0, 3, 3, 3, 6, \cdots\}$ which corresponds to when process 3 has a state transition.

Prediction proceeds in two steps. First, we obtain predictions $q(\tilde{h}_t^{(i)})$ from each component HMM. Second, in order to eliminate the persistent bias effect mentioned above, we estimate the noise terms $\tilde{\epsilon}_t^{(i)}$ for $i = 2 \ldots M$ and add these to $q(\tilde{h}_t^{(i)})$ to obtain an estimate of $q(h_t^{(i)}) + \epsilon_t^{(i)}$. We first estimate $\tilde{\epsilon}_{t+1}^{(M)}$, where $M$ is the slowest HMM process, as

$$\mathrm{E}[\tilde{\epsilon}_t^{(M)}|h_t, X_t, \cdots, X_{g^{(M)}(t)}] = \frac{1}{t - g^{(M)}(t)} \sum_{j=0}^{t - g^{(M)}(t)} X_{t-j} - q(\tilde{h}_t^{(M)})w^{(M)} \ ,$$

and subsequently estimate

$$\mathrm{E}[\tilde{\epsilon}_t^{(i)}|h_t, X_t, \cdots, X_{t-g^{(i)}(t)}] = \frac{1}{t - g^{(i)}(t)} \sum_{j=0}^{t-g^{(i)}(t)} X_{resid,t-j}^{(i)} - q(\tilde{h}_t^{(i)})w^{(i)} \ ,$$

and finally predict

$$X_{t+1} = \tilde{q}(h_t)w = \left[ q(\tilde{h}_t^{(1)}) + \tilde{\epsilon}_t^{(1)}, \cdots, q(\tilde{h}_t^{(M)}) + \tilde{\epsilon}_t^{(M)} \right] w.$$

This correction term estimation procedure must be repeated at all times $t$ for all component HMMs. Algorithm 1 details the prediction procedure.

---

**Algorithm 1** Prediction using MSHMM

---

1: **Input:** Observed emissions $X$.
2: Let $X_{resid}^M = X$, and set $i = M$.
3: **while** $i > 0$ **do**
4:     Predict $X_{t+1}^{(i)}$ using spectral estimation HMM, $q(\tilde{h}_t^{(i)})$ (see Rodu et al. (2013) for details) from the observations $X_{resid}^{(M)}$.
5:     Compute the error $X - q(\tilde{h}_t^{(i)})w^{(i)}$ over $(g^M(t), t]$.
6:     Estimate new $\tilde{\epsilon}_t^{(i)}$ by averaging the error over this period.
7:     Compute $X_{resid}^{(i-1)} = X_{resid}^{(i)} - q(\tilde{h}_t^{(i)})w^{(i)} - \tilde{\epsilon}_t^{(i)}$ and set $i = i - 1$.
8: **end while**
9: Sum over all predictions from each process $X_{t+1}^{(i)}$.

---

### 2.3 Estimation Algorithm for Multiscale Hidden Markov Models

We divide the algorithm into two components. The first estimates the HMM parameters through the spectral method of moments (see Rodu (2014) for details). The second, presented in algorithm 2, estimates the weight vector that combines the emissions of the component HMMs in the MSHMM.

The critical assumptions that allow our model to be estimated using fast and consistent spectral methods are 1) the independence of the component HMM processes and 2) the fact that the emissions $X_t$ are linear combinations of the latent emissions from the component HMM processes. Note that the decoupling of the HMMs is possible because they evolve at different rates.

---

**Algorithm 2** Computing weights for the linear combination of HMMs

---

1: **Input:** Training set of observed emissions $X$.
2: Let $X_{resid}^{(M)} = X$, and set $i = M$.
3: **while** $i > 0$ **do**
4:     Using spectral estimation method from Foster et al. (2012) (see supplementary materials), estimate the HMM process $i$ with a binned average $\left( X_{avg(\Delta^{(i)})}^{(i)} \right)$ of bin width $\Delta^{(i)}$ over time series $X_{resid}^{(i)}$, and compute observation expectations $q(h_t^{(i)})$.
5:     In order to recenter, compute $X_{resid}^{(i-1)} = X_{resid}^{(i)} - X_{avg(\Delta^{(i)})}^{(i)}$ and set $i = i - 1$.
6: **end while**
7: Estimate $w$ with equation $X_t = q(h_t)w + \epsilon_t$ with $q(h_t) = \left[ q(h_t^{(1)}), q(h_t^{(2)}), \ldots, q(h_t^{(M)}) \right]$.

---

### 2.4 Theoretical Foundations of Multiscale Hidden Markov Models

In our estimation scheme, observations for HMM process $M$ are obtained by segmenting the time series into bins of size $\Delta^{(M)}$. A residual time series is then constructed by subtracting the derived observations from their respective observations. We then obtain observations for HMM process $M - 1$ by binning the residual time series in a similar fashion, and proceed recursively for every other process. Each time we bin, however, there is noise in the estimated observation, and that noise gets propagated to faster chains. Critically, the only effect of this noise on consistency is bias in the

estimation of the *diagonal* elements of the *second* moment, which only appears in the method of moments estimation in its inverted form.

More technically, note that in our estimation scheme we assume that $E[X_t^{(i)}] = 0$ for $i < M$. In an MSHMM with only two processes, for instance, this implies that $E[X_t^{(1)}] = 0$. Let $q^{(1)}$ be a matrix such that

$$q^{(1)} h_t^{(1)} = q(h_t^{(1)}) = E[X_t^{(1)} | h_t^{(1)}]$$

then under stationarity conditions we have that $q^{(1)} \pi^{(1)} = 0$. This implies that $E[X_{t'}] = X_t^{(2)}$ for $t' \in [t, t + \Delta^{(2)})$, which suggests that we first estimate observations $X_{t+k\Delta^{(2)}}^{(2)}$ of the slow-moving HMM by calculating the expected value of $X_{t'}$ for $t' \in [t, t + \Delta^{(2)})$, then estimating the slow-moving model using these calculated observations and the fast-moving model using the residuals of observations from their expected value. But because we have only $S = \Delta^{(2)}/\Delta^{(1)}$ observations $X_{t'}$ for $t' \in [t, t+\Delta^{(2)})$ our estimate is noisy. The main problem is that the bias in the empirical estimate of $X_t^{(2)}$ is reflected in $X_{t'} - X_t^{(2)}$ for all $t' \in [t, t + \Delta^{(2)})$. This, in turn, biases the estimation of the diagonal elements of the second moment matrix of the faster HMM 1. It does not, however, bias estimation of HMM 2, as the error between two successive observations for that HMM are independent.

In practice, the bias seems to have no real effect on the performance of the MSHMM, and even might serve as a natural regularization. It is possible, however, to estimate the spectral parameters so as to eliminate this bias, which we lay out in theorem 1. This provides a practical mechanism for obtaining consistent estimates that effectively restricts the sample size of each component HMM process to be on the order of that of the slowest HMM. On the other hand, if we let the ratio between sampling intervals become arbitrarily large as in theorem 2, while impractical, this allows observations estimates from each HMM component to become arbitrarily precise, thus again providing consistent estimators.

**Theorem 1.** *Assume we have an MSHMM. Let $^k\Delta_i^{(M)}$ and $^k\Delta_i^{(M)} + 1$ be the last observation in HMM $k$ before the $i^{\text{th}}$ state transition of HMM $M$ and the first observation in HMM $k$ immediately following the $i^{\text{th}}$ transition of HMM $M$, respectively. Note that for HMM $k$ the emission value at time $t = \Delta_i^{(M)}$ is equal to the previous $\Delta^{(k)}$ time steps. Assume that the second order observables for all chains $k \in \{1, \ldots, M\}$ are estimated only using the bigram $^k\Delta_i^{(M)}$ and $^k\Delta_i^{(M)} + 1$. Then the estimated spectral MSHMM parameters converge to the true spectral MSHMM parameters.*

*Proof.* We consider the two layer process, though this generalizes to an arbitrary number of layers. As a reminder, we can decompose the theoretical contributions from the faster-moving HMM 1 into two components: the state-conditional means and a mean 0 noise term that we consider to be drawn iid at each time step. Similarly, empirical contributions can be decomposed as the state-conditional means, an iid noise term, and an estimation bias propagated from HMM 2 that persist for $\Delta^{(2)}$ time steps. Note that from the central limit theorem the bias terms are drawn iid with mean 0. Therefore when estimating the second order observable as suggested in theorem 1 the bias term simply acts as an additional noise term. It is straightforward to see, then, that all the necessary observables converge to their true parameters. ☐

**Theorem 2.** *Assume we have an MSHMM as defined above, and let $J$ be the total number of observations. Let $J \to \infty$ and $\Delta^{(i)}/\Delta^{(i-1)} \to \infty$ for $1 < i \leq M$. Then the estimated spectral MSHMM parameters converge to the true spectral MSHMM parameters (in the sense that the probability forecasts are consistently estimated.)*

*Proof.* Considering the two layer MSHMM, we show that as $\Delta^{(i)}/\Delta^{(i-1)} \to \infty, \hat{X}_t^{(2)} \to X_t^{(2)}$. We write

$$\hat{X}_t^{(2)} = \frac{1}{S} \sum_{t' \in [t, t+\Delta^{(2)})} X_{t'} = X_t^{(2)} + q^{(1)} \hat{\pi} + \frac{1}{S} \sum_{t' \in (t, t+\Delta^{(2)})} \epsilon_{t'}^{(1)}$$

From the law of large numbers, as $S \to \infty$, $\frac{1}{S} \sum_{t' \in (t, t+\Delta^{(2)})} \epsilon_{t'}^{(1)} \to 0$, and $\hat{\pi}^{(1)} \to \pi^{(1)}$, and since $q^{(1)} \pi^{(1)} = 0$ we have $\hat{X}_t^{(2)} \to X_t^{(2)}$. ☐

While theorem 2 is impractical, it shows that as the separation of time scales between HMM processes increases, the empirical estimates of the spectral parameters converge to their theoretical counterparts. If a comfortably large enough separation is not feasible or desirable, theorem 1 provides an estimation method that inherently enjoys the usual consistency of spectral HMM estimation. In practice, however, the effect of the bias is minimal.

## 3 EXPERIMENTS WITH SYNTHETIC DATA

We assessed the estimation speed and accuracy of MSHMMs using synthetic data generated from an MSHMM process containing 3 component HMMs each with 5 hidden states (MSHMM-3-5) which emits a vector of continuous observations. Computation time for training the MSHMM was linear in the number of observations and quadratic in the number of states. For our application, we found no benefit beyond 15 hidden states. Our computational times are consistent with Parikh et al. (2012). Furthermore, because of the MSHMM specification, the HMM with the highest resolution dominated the computational time for estimating the model. For comparison, a simple HMM with 5 hidden states using EM[1] required 1255 seconds to estimate parameters for 900,000 observations while our MSHMM-3-5 took 25 seconds.

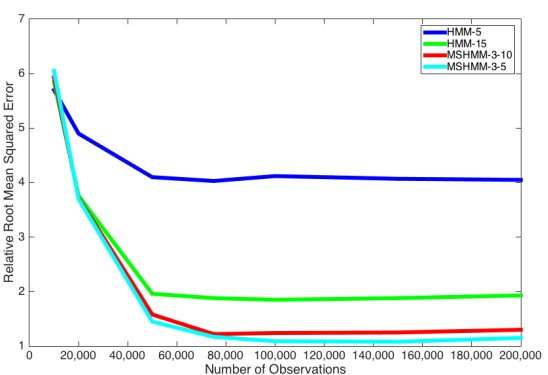

Figure 2: Model RMSE for MSHMM normalized by the RMSE of the oracle. Thus 1 is the absolute minimum.

In Figure 2 we show our results using synthetic data. We compared the correct MSHMM to HMMs with 5 and 15 hidden states. We also compared a 3 Markov component MSHMM which has 10 hidden states instead of 5. Finally, as an upper bound on performance we used the "oracle" MSHMM which makes prediction using the true hidden state transition, emission, and weight parameters, where only the hidden state sequence is unknown. Surprisingly, the MSHMM with too many hidden states still performs well. The relative root-mean-squared error (RMSE) is the RMSE normalized by the oracle RMSE. This points to the fact that the model is robust to some misspecification; however, further regularization might improve results as shown by Tran et al. (2016).

We also compared our model against Long-Short Term Memory (LSTM) (Hochreiter & Schmidhuber, 1997) recurrent networks, which with a 30 dimensional hidden state trained using all 200,000 data points gave a relative RMSE of 1.76. While this is much better than the results from HMM-15, the LSTM failed to capture multiscale long term dependencies and so underperformed our MSHMM. We further examined the State Frequency Memory (SFM) recurrent network (Hu & Qi, 2017) , which incorporates frequency information into the memory cells of an LSTM. Its relative RSME was 1.69 using 10 frequency banks, again giving poorer accuracy than the MSHMM

## 4 ESTIMATING STOCK PRICE COVARIANCE

### 4.1 DATA DESCRIPTION

Daily stock closing prices were extracted from the Center for Research in Securities Prices (CRSP) database.[2] All intraday data was extracted from the NYSE Trade and Quote (TAQ) database.[3] All stock returns are calculated using the difference of log prices. The calculation of realized covariance uses the same procedure as Sheppard & Xu (2014).

---

[1]http://cran.r-project.org/web/packages/HMMCont/HMMCont.pdf
[2]http://www.crsp.com/products/research-products/crsp-us-stock-databases
[3]http://www.nyxdata.com/doc/2549

We sampled the realized covariance at daily, 17-minute and 1-minute rates. We chose the 17-minute sample rate since Bollerslev et al. (2008) have shown that microstructure effects on realized variance are mostly mitigated. We chose six exchange-traded funds (ETFs), including SPY (which tracks the S&P 500 index), TLT, FEZ, USO, UNG, and GLD. Each realized covariance matrix has 21 unique entries. We calculated realized variance over 10 years of historical daily data, 2 years of 17-minute intraday data, and 1 year of 1-minute intraday data. To be exact, for training we used 2268 daily, $\sim$ 22000 17-minute, and $\sim$ 20000 1-minute observations, and for testing we used 252 daily, $\sim$ 2600 17-minute, and $\sim$ 4000 1-minute observations. Due to market conditions such as stock halts, the number of intraday data points is not the same for all stocks, so we reported approximate number of observations.

## 4.2 EMPIRICAL RESULTS

Our model was applied to high-frequency equity data for covariance estimation. We estimated the slowest process using 9 years of data, while the minute-level process was estimated using shorter periods of time. This is another useful feature of our model versus other hierarchical HMMs: we can estimate the daily covariance process using a longer overlapping time period.

We used the standard Generalized Autoregressive Conditional Heteroskedastic model (GARCH) (Bollerslev, 1986) as our baseline. We chose the baseline model to be GARCH(1,1), such that

$$\sigma_t = \sqrt{h_t}v_t,\ v_t \sim N(0,1)\ \text{and}\ h_t = \zeta + \alpha\sigma_{t-1}^2 + \gamma h_{t-1}.$$

The prediction from GARCH(1,1) is the estimate of volatility $\sigma_t^2$. GARCH is a standard baseline for volatility prediction. We also compared our model to two types of models. The first is a generalization of the multivariate GARCH, Bayesian Multivariate Dynamic Covariance model (BMDC)[4] (Wu et al., 2013). The second type of model is a generalization of regime-switching models to high-frequency data, High-Frequency-Based Volatility (HEAVY)[5] models (Noureldin et al., 2012; Sheppard & Xu, 2014). BDMC and HEAVY have been shown to be computationally efficient to estimate as well as to perform at state-of-the-art levels. The model seeks to predict the covariance either a day, 17 minutes, or a minute forward.

| Horizon | BDMC | HEAVY | LSTM-30 | SFM-30-10 | MSHMM-3-5 |
|---|---|---|---|---|---|
| daily | **0.39** ( 976 s) | 0.42 ( 103 s) | 0.78 | 0.73 | 0.93 ( **73 s**) |
| 17-minute | 0.77 ( 9134 s) | 0.76 ( 717 s) | 0.84 | 0.82 | **0.73** ( **51 s**) |
| 1-minute | 0.75 ( 9576 s) | 0.80 ( 1879 s) | 0.77 | 0.73 | **0.51** ( **47 s**) |

Table 1: The RMSE of the out of sample predicted versus realized stock covariance which is normalized by the GARCH(1,1) RMSE. Run times are in parenthesis in seconds. The training time for neural models are between 20 minutes and 5 hours using a single Titan X GPU and highly variable.

We report the RMSE of the models normalized with respect to the GARCH baseline. Table 1 shows the results of the prediction on ETFs. MSHMM is clearly superior at higher frequency, where the realized covariance matrix tends to be sparse. This result is consistent with our simulated data. When we used a daily prediction horizon, MSHMM underperformed BDMC, HEAVY, and LSTM models; this should be expected, however, since high-frequency data does not aid estimation very much when estimating the covariance only at the market close. MSHMM outperformed both models when predicting at 17-minute intervals and 1-minute realized covariance. MSHMM is also significantly faster than BDMC and HEAVY models [6]. MSHMM outperforms the LSTM and SFM neural models[7] in part due to the fact that there are daily jumps between the market close and subsequent open on the next trading day. We needed to remove the overnight return effect in order to improve upon GARCH(1,1) performance. Finally, we again draw the reader's attention to the fact that **without** the estimation of noise as done in algorithm 1, MSHMM underperformed even GARCH(1,1).

---

[4]Code is from `https://bitbucket.org/jmh233/bmdc_icml2013`

[5]Code is from `https://www.kevinsheppard.com/MFE_Toolbox`

[6]We used the highly optimized C++ code of HEAVY and the Matlab code from BDMC. Our model is written in R and not yet optimized for performance.

[7]Code is from
`https://github.com/z331565360/State-Frequency-Memory-stock-prediction`

## 5 RELATED WORK

The MSHMM is a specialized hierarchical HMM, in a similar vein proposed by El Hihi & Bengio (1995), and is distinct from the factorial hidden Markov models (FHMMs) of Ghahramani & Jordan (1997). While FHMMs and MSHMMs are both capable of learning short- and long-term structure, the MSHMM explicitly differentiates time-scale evolution.

**Spectral estimation of latent state models** Recently there has been significant interest in spectral estimation of latent state models, beginning with the seminal work in spectral estimation of discrete HMMs (Hsu et al., 2012) which has been extended to fully reduced methods (Foster et al., 2012), to continuous output HMMs (Song et al., 2010; Rodu et al., 2013), to trees, (Dhillon et al., 2012), and to more general latent variable structures (Anandkumar et al., 2014; Parikh et al., 2012). In this paper, we extend spectral estimation techniques to MSHMMs.

**Hierarchical HMM variants** A multitude of hierarchical HMM variants have been proposed since hierarchical HMMs were first introduced (El Hihi & Bengio, 1995). Hierarchical HMMs are characterized by multiple interconnected hidden states, $\Pr[h_{t+1} \mid h_t, h_t', \cdots]$ where $h_t'$ is not in the same hidden state sequence, or by multiple hidden states that affect an emission, $\Pr[x_t \mid h_t, h_t', \cdots]$. FHMMs are a subclass of hierarchical HMMs in which multiple independent hidden state processes affect an observable emission. There are a multitude of variants on FHMMs. Among these are multirate HMMs (Cetin & Ostendorf, 2004) and additive FHMMs (Kolter & Jaakkola, 2012), which differ in structure from our model.

Multirate HMMs have multiple hidden state processes, which we refer to as "levels" or "components," each evolving at different rates. The main structural difference between the multirate HMM and our model is that only the first level emits an observable; all subsequent "higher" levels output to the level "below," which has a faster hidden state process. MSHMM has additive emissions results, whereas, multirate HMM will propagate the same error upstream through different levels of hidden states.

By way of contrast, in additive FHMMs all hidden state processes evolve at the same rate, but the higher level process emits to both observations at $t$ and $t+1$. It is possible to formulate an MSHMM and more generally hierarchical HMM as a latent junction tree, on which spectral algorithms can be estimated (Parikh et al., 2012). However, latent junction trees are inefficient representations for estimating FHMMs and MSHMMs (Jordan et al., 1999) as they couple the hidden state paths, leading to an explosion in the number of parameters to be estimated.

**Neural Models** Deep neural networks allow a non-linear form of hierarchical HMMs as seen in the work of El Hihi & Bengio (1995) and more recently Liu et al. (2015); Chung et al. (2016). Similar to junction trees, these models require exponentially large hidden state representations (Jaderberg et al., 2016) compared to MSHMM. The work of Chung et al. (2016) is more similar to multi-rate HMMs than to MSHMM. Nonetheless, exploration of a neural equivalent to MSHMM would be interesting. The SFM recurrent networks (Hu & Qi, 2017; Zhang et al., 2017) have been shown to be useful for stock price prediction, which is a related task. However, SFM does not explicitly incorporate multiple time scales, which is critical for covariance estimation. Another recent neural model is the neural volitility model (Luo et al., 2017) which has shown superior results than GARCH models; however, this model is not multiscale.

**Financial Covariance Estimation Models** The standard methods for modeling stock price variation trade off between parameter accuracy and computational efficiency. The modern classes of models are multivariate versions of generalized autoregressive conditional heteroscedasticity (GARCH) (Engle & Kroner, 1995; Wu et al., 2013), stochastic volatility models (Gouriéroux et al., 2009), and Markov-switching processes (Calvet & Fisher, 2008), (which, while a hidden state model, their formulation and estimation are significantly different). Furrer & Bengtsson (2007) used a Kalman filter approach in order to estimate covariance matrices. More recently, Nystrup et al. (2016) modelled covariance using HMMs. Most of these models do not scale to large amounts of data and are used to predict daily returns or even longer periods. For a more detailed overview of models for large covariance matrix estimation models see Fan et al. (2016).

With the emergence of high-frequency data containing detailed records of quotes and transaction prices with nanosecond time resolution, a new class of models has emerged, such as multivariate high-frequency-based-models (HEAVY) (Noureldin et al., 2012). Our model extends the work of

Rossi & Gallo (2006) to multiple timescales, covariance estimation, and the use of computationally efficient spectral methods instead of EM.

## 6 CONCLUSION

Our MSHMM is a specialized hierarchical HMM that captures multiscale structure by adding emissions from multiple component HMMs. Unlike standard factorial HMMs, MSHMM handles noise at multiple time scales in a manner that cleanly supports spectral estimation. This estimation is straightforward if one accounts for the noise that propagates from the slower to the faster time scales. However, prediction requires accounting for the fact that estimation error due to noise at slower time scales acts as a bias term for the faster component HMMs. We have provided fast estimation schemes with provable asymptotic properties.

Because we cast MSHMMs in the spectral estimation framework, unlike other hierarchical HMMs, they scale gracefully with the size and dimensionality of the data, increasing the range of problems they can address. They apply naturally to high-frequency financial time series, as they capture both the short- and long-term covariances between stock prices, which are used as the basis for decision-making about portfolio management and risk assessment, and they rapidly detect market changes. The MSHMM can also potentially be applied to other types of multi-time scale data in domains such as natural language processing, vision, speech recognition, neuroscience, and macroeconomics.

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

# Supplemental Material

## A  SPECTRAL ESTIMATION ALGORITHM

---

**Algorithm 3** Computing observables for spectral estimation of an HMM, fully reduced third moment

---

1: **Input:** Training examples- $\{x_{1_i}, x_{2_i}, x_{3_i}\}$ for $i \in \{1, \ldots, NT\}$.
2: Compute $\hat{E}[x_2 \otimes x_1] = \frac{1}{NT} \sum_{i=1}^{NT} x_{2_i} x_{1_i}^\top$.
3: Compute[8] the left $k$ singular vectors corresponding to the top $k$ singular values of $\hat{E}[x_2 \otimes x_1]$. Call the matrix of these singular vectors $\hat{U}$.
4: Reduce data: $\hat{y} = \hat{U}^\top x$.
5: Compute $\hat{\mu} = \frac{1}{NT} \sum_{i=1}^{NT} y_{1_i}$, $\hat{\Sigma} = \frac{1}{NT} \sum_{i=1}^{NT} y_{2_i} y_{1_i}^\top$ and tensor $\hat{C} = \frac{1}{NT} \sum_{i=1}^{NT} y_{3_i} \otimes y_{1_i} \otimes y_{2_i}$.
6: Set $\hat{b}_1 = \hat{\mu}$ and $\hat{b}_\infty^\top = b_1^\top \hat{\Sigma}^{-1}$.
7: Right multiply each slice of the tensor in the $y_2$ direction (so $y_2$ is being sliced up, leaving the $y_3 y_1^\top$ matrices intact) by $\hat{\Sigma}^{-1}$ to form $\hat{B}(\gamma) = \hat{C}(\gamma)\hat{\Sigma}^{-1}$.

---

Spectral estimation can be understood through the lens of the observable operator model (Jaeger, 2000). For simplicity of presentation we describe spectral estimation for a discrete emission HMM. Consider an HMM where $T$ is an $m \times m$ transition matrix on the hidden state, $O$ is a $v \times m$ emission matrix giving the probabilities of hidden state $h = j$ emitting observation $x = i$, and $\pi$ is a vector of initial state probabilities in which $\pi_i$ is the probability that $h_1 = i$. Jaeger Jaeger (2000) showed that the joint probability of a sequence of observations from this HMM is given by

$$Pr(x_1, x_2, \ldots, x_t) = 1^\top A_{x_t} A_{x_{t-1}} \cdots A_{x_1} \pi,$$

where $A_x \equiv T \text{diag}(O^\top \delta_x)$, $\delta_x$ is the unit vector of length $v$ with a single 1 in the $x$th position and $\text{diag}(v)$ creates a matrix with the elements of the vector $v$ on its diagonal and zeros everywhere else. $A_x$ is called an 'observation operator'.

Now, define a random variable $y_t = U^\top \delta_{x_t}$, where $U$ has orthonormal columns and is a matrix mapping from observations to the reduced dimension space, then

$$Pr(x_1, x_2, \ldots, x_t) = b_\infty^\top B_{y_t} B_{y_{t-1}} \cdots B_{y_1} b_1$$

holds where

$$
\begin{aligned}
b_1 &= \mu \\
b_\infty^\top &= \mu^\top \Sigma^{-1} \\
B_y &= C(y) \Sigma^{-1}
\end{aligned}
$$

and $\mu = E(y_1)$, $\Sigma = E(y_2 y_1^\top)$, and $C(y) = E(y_3 y_1^\top y_2^\top)y$. Writing $\mu$, $\Sigma$, and $C(y)$ in terms of their theoretical quantities, we get

$$
\begin{aligned}
\mu &= U^\top O \pi \\
\Sigma &= U^\top O \, T \, \text{diag}(\pi) \, O^\top U \\
\Sigma^{-1} &= (O^\top U)^{-1} \, \text{diag}(\pi)^{-1} \, T^{-1} \, (U^\top O)^{-1} \\
C(y) &= U^\top O \, T \, \text{diag}(O^\top U y) \, T \, \text{diag}(\pi) \, O^\top U
\end{aligned}
$$

and thus can be shown that

$$
\begin{aligned}
b_1 &= U^\top O \, \pi \\
b_\infty^\top &= \mathbf{1}^\top (U^\top O)^{-1} \\
B_y &= U^\top O \, A_x \, (U^\top O)^{-1}
\end{aligned}
$$

and thus

$$
\begin{aligned}
Pr(x_1, x_2, \ldots, x_t) &= 1^\top A_{x_t} A_{x_{t-1}} \cdots A_{x_1} \pi \\
&= b_\infty^\top B_{y_t} B_{y_{t-1}} \cdots B_{y_1} b_1
\end{aligned}
$$

## B    COMPARISON TO RELATED HIDDEN MARKOV MODEL

Multirate HMMs have multiple hidden state processes, which we refer to as "levels" or "components," each evolving at different rates. The main structural difference between the multirate HMM (as seen in figure 1) and our model (figure 3) is that only the first level emits an observable; all subsequent "higher" levels output to the level "below," which has a faster hidden state process. MSHMM has additive emissions results, whereas, multirate HMM will propagate the same error upstream through different levels of hidden states.

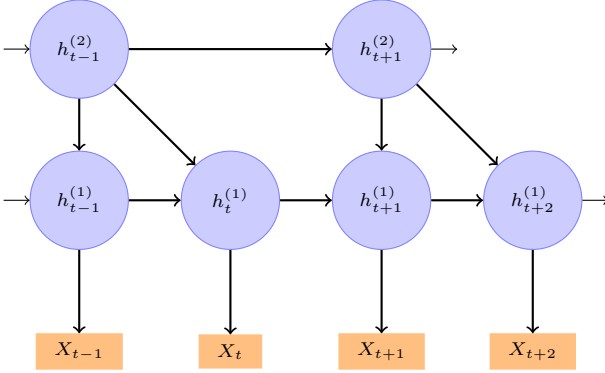

Figure 3: Multi-rate HMM with 2 hidden state processes.

By way of contrast, in additive FHMMs (figure 2) all hidden state processes evolve at the same rate, but the higher level process emits to both observations at $t$ and $t + 1$.

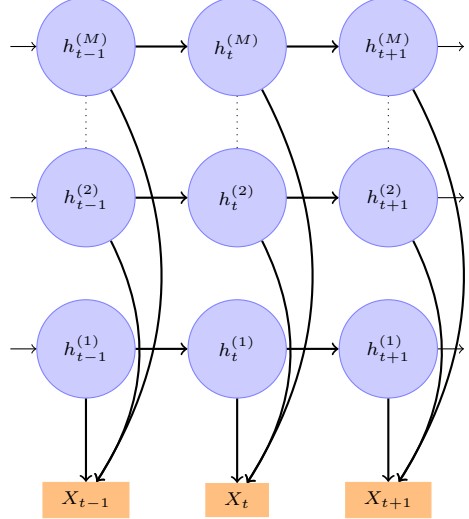

Figure 4: Factorial HMM with M hidden state processes.

## C    SYNTHETIC DATA

The first 2500 observations for the first entry in the emission vector are plotted in figure 4. One can see the effect of the transitions of different component HMMs.

The computational time for both training and prediction of the MSHMM is shown in figure 5.

## D    FURTHER DISCUSSION OF OTHER HIERARCHICAL HMMS

Further, latent junction trees are constrained to a fixed tree topology. This has two ramifications. First, a separate model needs to be estimated for each desired topology, and more importantly, in the application presented here,

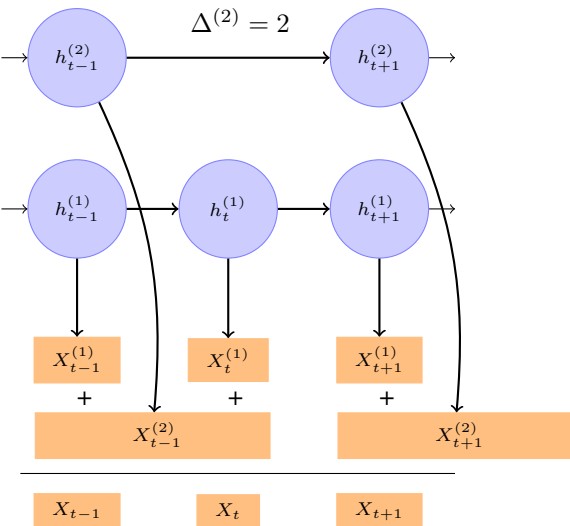

Figure 5: An MSHMM with two HMMs where $h_t^{(2)}$ is the hidden state sequence of the slower moving HMM with transitions happening at time steps $\Delta^{(2)} = 2$. The observations $X_t$ are the sum of the emissions of the two HMMs, so $X_t = X_t^{(1)} + X_{t-1}^{(2)}$.

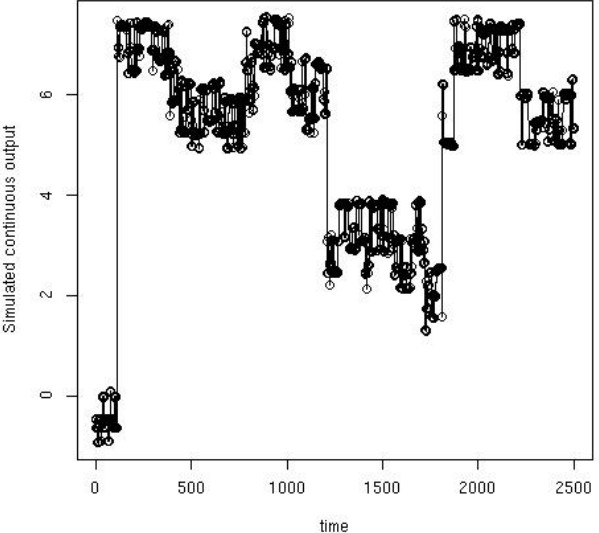

Figure 6: Synthetic MSHMM data plot for 3 hidden state level and 5 hidden states for each (MSHMM-3-5)

the tree topology would have to be extremely wide to accommodate the desired length of observation history, requiring an unmanageable computational burden.

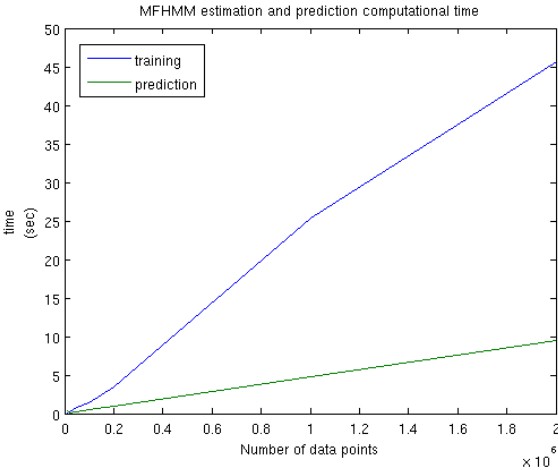

Figure 7: Computational time for MSHMM

# E    FURTHER DETAILS OF DATA

Daily stock closing prices were extracted from the Center for Research in Securities Prices (CRSP) database[9]. Daily returns where adjusted for corporate actions such as dividends and stock splits. All intraday data was extracted from the NYSE Trade and Quote (TAQ) database[10]. Intraday prices where filtered using trade flags and incorrect price quotations. Our reference intraday prices used the average of the best bid price and best ask price, and all stock returns are calculated using the difference of log prices.

Figure 6 is illustrative of the multiple scales of stock variance of SPY. In particular, note that there are clearly different states for daily and intraday realized variance.

We chose the 30 stocks: TROW, AAPL, AA, CMA, FITB, BEN, MMC, HRB, BK, AFL, AXP, CB, IBM, MSFT, PNC, XOM, PGR, KEY, LM, SLM, AIG, STI, ALL, COF, UNH, SPY, GLD, TLT, FEZ, USO, UNG , as well as exchange trader funds (ETFs) such as SPY, which tracks the S&P 500 index, and GLD which tracks the value of gold prices. Each realized covariance matrix has 465 unique entries. We calculated realized variance over 10 years of historical daily data, 2 years of 17 minute intraday data, and 1 year of minute intraday data. To be exact, for training we used 2268 daily, $\sim$ 22000 17 minute, and $\sim$ 20000 minute datapoints, and for testing we used 252 daily, $\sim$ 2600 17 minute, and $\sim$ 4000 minute datapoints. Due to market conditions such as stock halts, the number of intraday data points is not the same for all stocks, so we reported approximate number of observations. Particularly, the top subplot of figure 7 corresponds to the adjusted daily prices of SPY and GLD.

Figure 7 also demonstrates the covariance between SPY and GLD, illustrating the multiscale nature of the covariance between SPY and GLD.

# F    FURTHER DISCUSSION OF EVALUATION

In the case of covariance prediction we define $MSE = \frac{1}{T-k} \sum_1^{T-k} (C - E(C)))^2$.

A close analysis of the results shows that HEAVY performs better at estimating covariance of stocks but not ETFs (SPY, GLD, USO, etc ). We surmised that this is because the covariance matrix for stocks tends to be close to a diagonal matrix, unlike ETFs, which have more significant off diagonal components. Since the variance is larger than the covariance, this accounted for the outperformance. We surmised that this is because companies are driven by very few factors, as a result the MSHMM hidden states focused on predicting covariance of ETF. However, at relatively high frequency timefames, our model outperformed. Again the intuition is that microstructure creates both fat tails as well as sparse covariance matrices, which are akin to discrete problems.

---

[9]http://www.crsp.com/products/research-products/crsp-us-stock-databases
[10]http://www.nyxdata.com/doc/2549

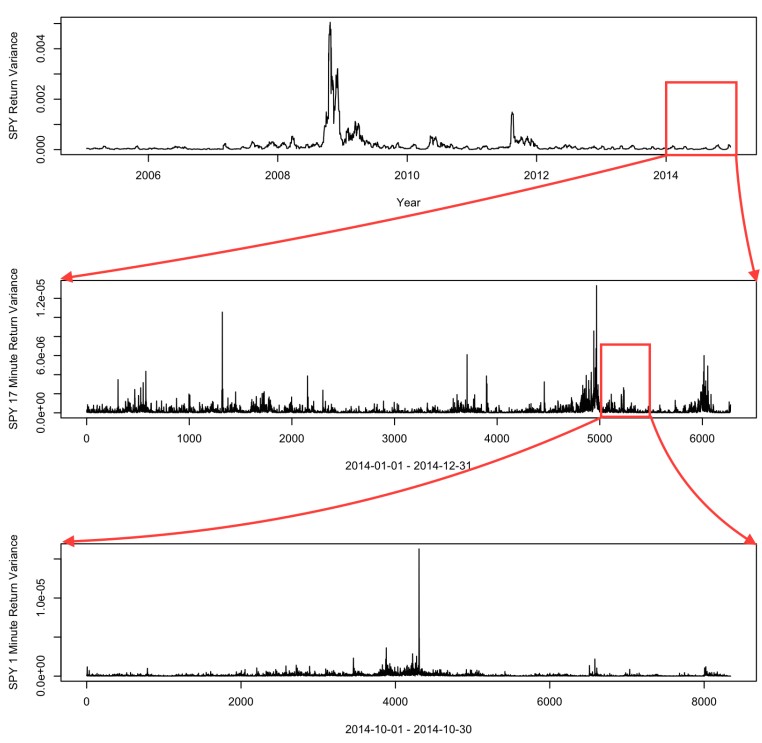

Figure 8: S&P 500 ETF (SPY) realized variance at daily, 17-minute, and 1-minute sampling rates over 10 years, 1 year, and 1 month respectively.

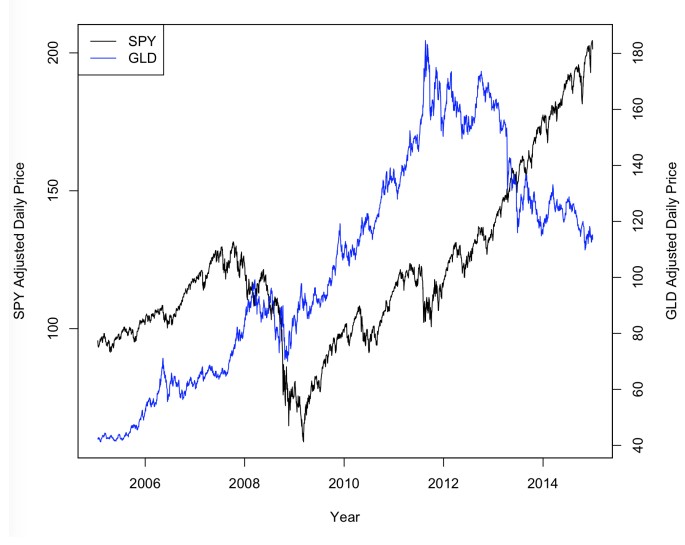

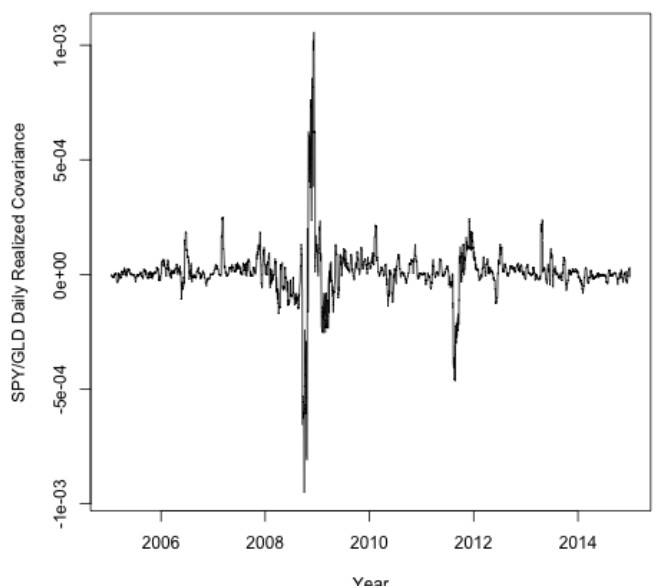

Figure 9: S&P 500 ETF (SPY) and SPDR Gold Shares ETF (GLD) daily prices, and their realized daily covariance.

