# OpenReview forum: "Multiscale Hidden Markov Models For Covariance Prediction"
_ICLR.cc/2018/Conference — Reject_

### Official Review · AnonReviewer2 · 2017-11-26
**Potentially useful approach for multi-scale Hidden Markov modelling but requires improvements**

**Rating:** 5
**Confidence:** 3

**Review:**

The paper focuses on a very particular HMM structure which involves multiple, independent HMMs. Each HMM emits an unobserved output with an explicit duration period. This explicit duration modelling captures multiple scale of temporal resolution. The actual observations are a weighted linear combination of the emissions from each latent HMM. The structure allows for fast inference using a spectral approach.

I found the paper unclear and lacking in detail in several key aspects:

1. It is unclear to me from Algorithm 2 how the weight vectors w are estimated. This is not adequately explained in the section on estimation.

2. The authors make the assumption that each HMM injects noise into the unobserved output which then gets propagated into the overall observation. What are reasons for his choice of model over a simpler model where the output of each HMM is uncorrupted?

3. The simulation example does not really demonstrate the ability of the MSHMM to do anything other than recover structure from data simulated under an MSHMM. It would be more interesting to apply to data simulated under non-Markovian or other setups that would enable richer frequency structures to be included and the ability of MSHMM to capture these.

4. The real data experiments shows some improvements in predictive accuracy with fast inference. However, the authors do not give a sufficiently broad exploration of the representations learnt by the model which allows us to understand the regimes in which the model would be advantageous.

Overall, the paper presents an interesting approach but the work lacks maturity. Furthermore, simulation and real data examples to explore the properties and utility of the method are required.

---

> ### Author Response · Authors · 2018-01-05
> **author response to reviewer 2**
>
> Thank you for your comments. We respond to the questions below.
>
> Regarding point 1. In our work we use linear regression. We will specify this in the paper. However, other methods can also easily be used to estimate w.
>
> Regarding point 2: From the perspective of the application, the slower time horizons are not deterministic, so we feel that this model better reflects the underlying data generation process. In the continuous emission HMM there is a noise term, so this was natural both from the application perspective and the model perspective.
>
> Regarding point 3: For other types of problems other than covariance prediction, model misspecification is more important.  We didn’t intend to do that in this work. We look forward to understanding model misspecification in future work.
>
> Regarding point 4: HMMs are widely used even though there is model misspecification. In our simulation we evaluated model misspecification, and in practice it is possible to assess the models predictive value by examining the realized error. For covariance prediction, we do not believe that the true process is a MSHMM, but our model is sufficient to predict the data.
> Can you provide further guidance as to which “regimes in which the model would be advantageous” would be interesting to test?
> We believe that MSHMM is useful in the class of problems where there are multiple processes and the ratio of \delta^{(i)} and \delta^{(i+1)} is sufficiently small that one cannot detrend and sufficiently large that a single HMM or LSTM model is insufficient. Problems such as
>
> We will run more synthetic data where the data generating process is only from the slowest HMM process and another only from the fastest process.

---

### Official Review · AnonReviewer1 · 2017-11-27

**Rating:** 6
**Confidence:** 4

**Review:**

This paper proposes a variant of hierarchical hidden Markov Models (HMMs) where the chains operate at different time-scales with an associate d spectral estimation procedure that is computationally efficient.

The model is applied to artificially generated data and to high-frequency equity data showing promising results.

The proposed model and method are reasonably original and novel.

The paper is well written and the method reasonably well explained (I would add an explanation of the spectral estimation in the Appendix, rather than just citing Rodu et al. 2013).

Additional experimental results would make it a stronger paper.

It would be great if the authors could include the code that implements the model.

---

> ### Author Response · Authors · 2018-01-05
> **author response to reviewer1**
>
> Thank you for your comments, we will certainly release the code upon acceptance of the paper.
>
> We have added further explanation of Rodu et al. 2013, in the Appendix and post an update version of the paper.
>
> We are definitely open to running further experiments. We could try synthetic data where the data generating process is only from the slowest HMM process and another only from the fastest process.
>
> In our update to the paper, we have compared our results to both LSTM and the very recent work State Frequency Memory (SFM) recurrent network [Hu and Qi, 2017].

---

### Official Review · AnonReviewer3 · 2017-12-04
**spectral algorithm for multiscale hmm, however it is not clear whether the algorithm is practically useful**

**Rating:** 6
**Confidence:** 4

**Review:**

The paper presents an interesting spectral algorithm for multiscale hmm. The derivation and analysis seems correct. However, it is well-known that spectral algorithm is not robust to model mis-specification. It is not clear whether the proposed algorithm will be useful in practice. How will the method compare to EM algorithms and neural network based approaches?

---

> ### Author Response · Authors · 2018-01-05
> **response to reviewer3**
>
> Thank you for your review and comments.
>
> Is it possible to provide a citation regarding the instability issue with model mis-specification?
>
> Regarding model robustness, as shown in [Tran et al., 2016, “Spectral M-estimation with Applications to Hidden Markov Models”] with sufficient data the misspecification does not produce significantly worse results. From both our simulated experiments as well as results for covariance prediction on real data, we believe that the model misspecification is not an issue for this type of problem. It is possible that regularization of the spectral algorithm would lead to more robust results; however, we leave this for future work.
>
> We believe that MSHMM is useful in the class of problems where there are multiple processes and the ratio of \delta^{(i)} and \delta^{(i+1)} is sufficiently small that one cannot detrend and sufficiently large that a single HMM or LSTM model is insufficient.
>
> EM is prohibitively slow for all but the daily covariance estimation, and thus excluded from analysis. As stated in paper, “For comparison, a simple HMM with 5 hidden states using EM required 1255 seconds to estimate parameters for 900,000 observations while our MSHMM-3-5 took 25 seconds.”
>
> We ran experiments to assess the LSTM performance.
> The synthetic data the relative RMSE was 1.76, which while worse than the MSHMM-3-5 and MSHMM-3-10, is better than both MSHMM-3-3 and HMM-15. It is expected that it should be better than HMM-15, but interesting that the LSTM exceeds MSHMM-3-3.
> Experiments with LSTM and an extremely recent variant SFM. We found that MSHMM outperforms the LSTM.
>
> We are interested in suggestions on further synthetic data experiments. We changed the data generating process is only from the slowest HMM process and another only from the fastest process. The performance is only slightly worse than the single HMM. Furthermore, the other HMM processes yield nearly 0 load in the regression.

---

### Decision · Program_Chairs · 2018-01-29
**ICLR 2018 Conference Acceptance Decision**

**Decision:**

Reject

**Comment:**

The paper addresses and interesting problem, but the reviewers found that the paper is not as strong as it could be: improving the range of evaluated data (significantly improve the convincingness of the experiments, and clearly adressing any alternatives, their limitations and as baselines).